# Influence of Waste Plastic Aggregate and Water-Repellent Additive on the Properties of Lightweight Magnesium Oxychloride Cement Composite

**Martina Záleská [1], Milena Pavlíková [1], Ondřej Jankovský [2], Michal Lojka [2], Filip Antončík [2], Adam Pivák [1] and Zbyšek Pavlík [1,*]**

[1] Department of Materials Engineering and Chemistry, Faculty of Civil Engineering, Czech Technical University in Prague, 166 36 Prague, Czech Republic; martina.zaleska@fsv.cvut.cz (M.Z.); milena.pavlikova@fsv.cvut.cz (M.P.); adam.pivak@fsv.cvut.cz (A.P.)

[2] Department of Inorganic Chemistry, Faculty of Chemical Technology, University of Chemistry and Technology, 166 28 Prague, Czech Republic; ondrej.jankovsky@vscht.cz (O.J.); michal.lojka@vscht.cz (M.L.); Filip.Antoncik@vscht.cz (F.A.)

* Correspondence: pavlikz@fsv.cvut.cz; Tel.: +420-22044-2002



**Featured Application: The results obtained in the paper can find use in the design and development of low-energy and low-carbon construction composites with the incorporation of waste expanded polypropylene. The developed "green materials" possess good thermal insulation function, minimum water absorption, sufficient permeability for water vapor, and resistance against the harmful water action. The mechanical strength of the lightened composites is acceptable for non-bearing purposes but if required, it can be further improved to meet the technical requirements of construction practice.**

**Abstract:** The aim of the present study is to improve the thermal and hygric performance of magnesium oxychloride (MOC) cement composites by the incorporation of waste plastic-based aggregate and the use of the inner and surface hydrophobic agents. The crushed waste expanded polypropylene particles were used as a full replacement of natural silica sand. The aggregate properties were evaluated in terms of their physical and thermal parameters. The caustic calcined magnesite was studied by SEM, XRF, and XRD spectroscopy. The MOC cement composites were characterized by SEM/EDS, XRD, and FT-IR spectroscopy and measurement of their structural properties, strength parameters, thermal conductivity, and volumetric heat capacity. Assessment of water- and water vapor transport properties was also conducted. The results show significantly improved thermal parameters of MOC cement composite containing expanded polypropylene (EPP) as aggregate and indicate high efficiency of surface hydrophobic agent (impregnation) as a barrier against the transport of liquid and gaseous moisture. The resulting lightweight EPP-MOC cement composite with improved thermal insulation function and suitable mechanical properties can be used to produce thermal insulation floors, ceilings, or wall panels reducing the operational energy demand of buildings.

**Keywords:** magnesium oxychloride cement; waste expanded polystyrene; mechanical properties; thermo-physical parameters; hygric properties; hydrophobic additives; water resistance

## 1. Introduction

Increasing concentrations of global greenhouse gas (GHG) emissions (especially $CO_2$), worldwide energy use, and amount of waste constitute currently the largest environmental problems. A large

volume of $CO_2$ emissions can be attributed to building industry, whether related to construction materials production or energy needed for heating and cooling of insufficiently insulated buildings [1,2]. For example production of Portland cement (PC) contributes to circa 5–7% of the global $CO_2$ emissions [3,4].

At present, there are efforts to find an alternative to PC that would have a low carbon footprint. One of the candidates may be cements based on the magnesium oxide (MgO). Magnesium oxychloride cement (also known as Sorel cement) is formed by the reaction of a light-burnt MgO powder with a solution of magnesium chloride ($MgCl_2$) [5], where the light-burnt (also called caustic c) MgO is produced by calcination of magnesite ($MgCO_3$) at a temperature of circa 750 °C [6]. This lower temperature, compared to ~1450 °C used in the production of Portland clinker, allows the use of alternative fuels [3]. MOC cement-based composites are able to absorb high amount of $CO_2$ from the atmosphere during their service life to form carbonates and hydroxycarbonates, which leads to their denser microstructure and to higher strength of resulting material [4,7]. The production of MgO from magnesite releases more $CO_2$ per ton than in the production of Portland cement (1.7 t of $CO_2$/t of MgO vs. 1 t $CO_2$/t of PC) [3]. However, taking into account the carbonation of the MOC cement composites, when MOC is able to sequestrate more than half of the $CO_2$ produced during $MgCO_3$ calcination (the resulting $CO_2$ emissions decreased therefore to around 0.5–0.6 t of $CO_2$/t), and given the total environmental impact during the life cycle of MgO ascertained by LCA, the MOC cement can be considered as environmentally friendly material [3].

Among superior properties of MOC cement composites are: early compressive strength, excellent fire resistance, low alkalinity (pH of 8–10), short setting time, and anti-abrasion performance [8,9]. MOC cements are also known by their ability to incorporate a higher amount of various fillers compared to PC, namely granite waste [10], fly ash [10,11], cenospheres derived from the fly ash [12], biomass ash [13], wood [6], recycled tire rubber [14], and waste plastics [15]. Currently, MOC cement is used for the production of industrial floors, for ornamental applications, stucco, grinding wheels, and for different types of panels used for fire protection, as decoration or for sound and thermal insulation [7,16]. As an air-dried cementing material, MOC cement composites lose their compressive strength after immersion in water because of the decomposition of hydration products. This limits their wider use in construction sector [9,17]. In literature, there are studies that report on improving the poor water resistance of MOC cement. Authors investigated the use of additives as soluble phosphates [18], fly ash [11,17,19,20], rice husk ash with addition of phosphoric acid, calcium lignosulfonate, and acrylic emulsion [21], the phosphoric acid [22,23], silica fudehume [19], and glass powder [21]. The evaluation of the MOC cement composites water-resistance is to date done only by the compressive strength measurement before and after the immersion in water. However, in the building practice it is also necessary to know other important and for specific applications even crucial parameters of construction materials, such as the water- and water vapor transport properties, heat transport, and storage properties, etc.

As a result of plastics versatility, durability, light-weight, and other intrinsic properties, the plastics demand is still increasing worldwide. The world plastic production was 359 million tons in 2018 and the European production constituted of approx. 17% of this amount [24]. As regards the demand for plastics in terms of the type of resin in Europe in 2017, on the first place was polypropylene (PP) with the 19.3%, followed by the low-density polyethylene (LD-PE) with 17.5% [25]. Among the foamed polymers, expanded polypropylene (EPP) has one of the largest worldwide productions. It is known particularly for its impact absorption, thermal insulation parameters, and a high strength to weight ratio. EPP has homogenous, closed cell structure and it is characterized by the good chemical and heat resistance. Unlike the foam polystyrene, it does not emit toxic gases when burned. The most common applications of EPP include packaging, automotive and industrial segments and safety components. The thermal insulation or acoustic properties of EPP are used in the construction of flooring parts in the building industry [26,27].

Lahtela et al. [28] reported on PE and PP polymers as the most common plastics in waste streams coming from construction and demolition waste and from mechanically separated plastic waste from a

sorting plant respectively. As the low biodegradability is also one of the properties of plastics, it is necessary to devise suitable end-of-life options for large quantities of plastics waste. Among the extensively studied recycling pathways are the incorporation of plastics waste particles as aggregate into Portland cement-based construction materials, which generally led to decrease of the unit weight and to improvement of the thermal insulation performance of the resulting material, but it is also associated with the reduction of the strength parameters [29–32]. The research on the replacement of natural silica aggregate in MOC cement composites with plastics waste aggregate reported, e.g., by Záleská et al. [15] is rather rare.

To date, only few studies about the thermal performance of MOC cement composites were published. Xu et al. (2016) reported on the decrease of the thermal conductivity of MOC mortars with the addition of cenospheres. Zgueb et al. [33] observed the low thermal conductivity of MOC cement blended with the polyvinyl acetate polymer (PVAc).

With respect to the lack of knowledge about the application of waste plastics as aggregate in MOC cement concrete, this study is focused on the assessment of MOC cement composites containing waste expanded polypropylene (EPP) particles as a full replacement of silica sand. For the improvement of water-resistance parameters, the inner (calcium stearate and sodium oleate) and surface (boiled linseed oil) hydrophobic agents were used. The produced materials were investigated in terms of their mineralogical composition, structural, mechanical, thermal, water-, and water vapor transport properties. The main aim was to develop lightweight environmentally friendly material with a reduced $CO_2$ emission footprint, characterized by the enhanced thermal insulation parameters and resistance to water penetration, which could be potentially used as a part of floors, ceilings or wall panels.

## 2. Experimental Section

### 2.1. Materials

The raw materials used in this paper were light-burned MgO powder, $MgCl_2 \cdot 6H_2O$, silica sand, and waste EPP particles. Crushed waste expanded polypropylene came from the aircraft models production. EPP was chosen for the investigation particularly with respect to its thermal and physical properties. In order to reduce the thermal conductivity of MOC cement composites as much as possible and with the knowledge of the high MOC cement binding capacity, it was decided to replace all natural aggregate. Based on the preliminary workability tests of the fresh mixtures [16], for further study the amount of the EPP particles of 150% by volume of natural silica sand was chosen. Magnesium oxide used in this paper was obtained from Styromagnesit Steirische Magnesitindustrie Ltd., Oberdorf, Austria and its chemical composition determined by X-ray fluorescence (EDXRF Spectrometer, ARL QUANT'X, Thermo Fisher Scientific, Waltham, MA, USA) is given in Table 1.

**Table 1.** Chemical composition of light-burned MgO powder obtained by XRF.

| Substance | Mass% |
|-----------|-------|
| MgO | 78.6 |
| CaO | 5.7 |
| $Fe_2O_3$ | 4.1 |
| $SiO_2$ | 4.4 |
| $Al_2O_3$ | 6.8 |

The main physical parameters of MgO powder such as specific density, powder density, and Blain specific surface are summarized in Table 2. The specific density was measured on a helium pycnometry principle using a Pycnomatic ATC (Porotec, Hofheim, Germany). This device is equipped with the real multi volume density analyzer and with the fully integrated temperature control with a precision of ± 0.01 °C. The powder density was calculated from the dry mass of the sample and its volume and the Blain specific surface was accessed in accordance with the standard EN 196-6 [34].

**Table 2.** Physical properties of light-burned MgO powder.

| Specific Density (kg·m$^{-3}$) | 3339 |
| --- | --- |
| Powder Density (kg·m$^{-3}$) | 838 |
| Blain Specific Surface (m$^2$·kg$^{-1}$) | 698 |

The X-ray diffraction analysis of MgO powder was examined at room temperature using Bruker D8 Discover (Bruker, Germany) powder diffractometer with parafocusing Bragg–Brentano geometry using CuK$_\alpha$ radiation (λ = 0.15418 nm, U = 40 kV, I = 40 mA). Data were scanned over the angular range 5–80° (2θ) with a step size of 0.019° (2θ). Results are shown in Figure 1. In addition to the main phase (magnesia), also some impurities were identified such as calcite, dolomite, and talc. This was in good agreement to XRF data.

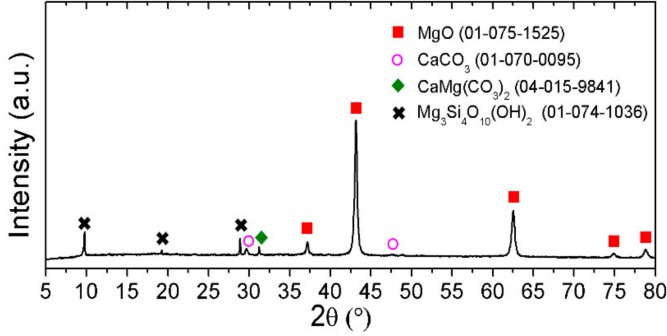

**Figure 1.** X-ray diffraction pattern of MgO powder.

An average particle size ($d_{50}$) of used MgO powder measured on a laser diffraction principle using an Annalysette 22 Micro Tec plus (FRITSCH, Idar-Oberstein, Germany) was approximately 45 μm. The cumulative and frequency distributions of particle size distribution are for the studied MgO powder introduced in Figure 2. The displayed data were acquired based on three independent measurements.

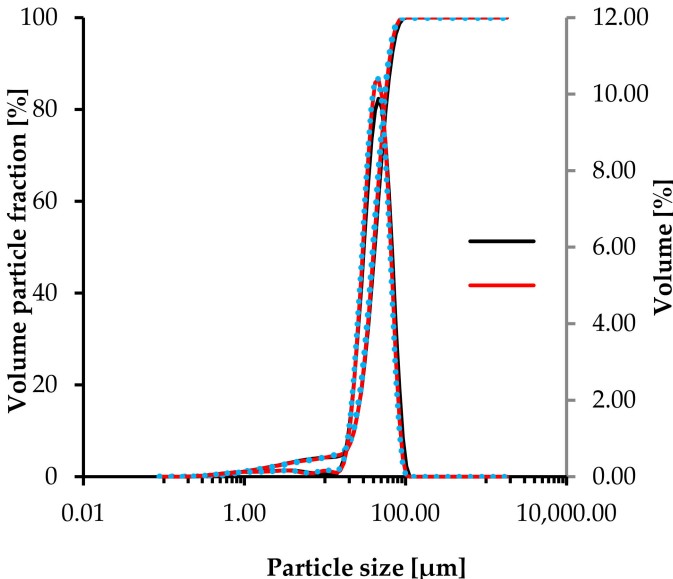

**Figure 2.** Particle size distribution of MgO powder—curative and frequency curves.

To prepare the magnesium chloride solution, $MgCl_2 \cdot 6H_2O$ of p.a. purity (Lach-Ner Ltd., Neratovice, Czech Republic) was dissolved in the tap water. For our solution we used 120.4 g of $MgCl_2 \cdot 6H_2O$ and 100 g of water.

The natural aggregate used for preparation of reference MOC cement composites was silica sand of particle size fraction 0–2 mm and specific density 2649 kg·m$^{-3}$. It was provided by Filtrační písky Ltd., Chlum u Doks, Czech Republic.

## 2.2. Sample Preparation

The mix preparation was carried out according to the standard EN 14016-2 [35]. The reference MOC cement composites (R-MOC) were prepared from MgO powder, $MgCl_2$ solution, and silica sand. MgO powder, $MgCl_2$ solution, and EPP aggregate were used for the production of composite-labelled EPP-MOC. Mixture with EPP aggregate and inner hydrophobic admixtures was named EPP-MOC-IH, as inner hydrophobic agents were used Ligastar CA 800 (calcium stearate) and Ligaphob N 90 (sodium oleate). These additives were provided by Excel Mix Cz, Ltd., Velim, Czech Republic. The hydrophobic admixtures were applied in the amount of 1 g of stearate and 2 g of oleate to 100 g of MgO powder. Stearate in the material reacts with water to form a gel that seals the structure and prevents moisture penetration. The interaction between the MgO powder and sodium oleate molecules can be expressed as van der Waals forces and hydrophobic forces [36].

The mixture proportions of MOC cement composites are summarized in Table 3. The fresh mixtures were casted in 40 mm × 40 mm × 160 mm prism molds, 70 mm × 70 mm × 70 mm cubic molds and in a circular mold having a diameter of 100 mm and a height of 20 mm. Samples were immediately after casting covered with polyethylene sheets, after 24 h demolded and then air-cured for next 27 days at a temperature of 23 ± 2 °C and relative humidity of 45 ± 5%. Examples of prepared cubic samples are displayed in Figure 3.

**Table 3.** Composition of MOC cement composite mixtures.

| | Mass (g) | | | | | |
|---|---|---|---|---|---|---|
| Mixture | Caustic Magnesite | MgCl$_2$ Solution | Sand | EPP | Calcium Stearate | Sodium Oleate |
| R-MOC | 450 | 500 | 1350 | – | – | – |
| EPP-MOC | 450 | 500 | – | 21.8 | – | – |
| EPP-MOC-IH | 450 | 500 | – | 21.8 | 4.5 | 9 |

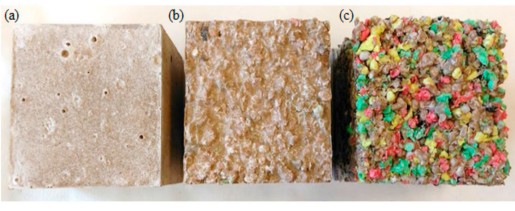

**Figure 3.** Tested samples: R-MOC (**a**); EPP-MOC-IH (**b**); EPP-MOC (**c**).

The R-MOC and EPP-MOC specimens intended for the measurement of water- and water vapor transport properties were coated with boiled linseed oil, which might affect the resulting diffusion and sorption properties of the material. The boiled linseed oil is used in the construction industry as a protective layer for wood and concrete elements, plasters, etc. It forms a thin layer on the concrete surface and clogs its pores, avoiding the penetration of water and chemical solutions into the pore structure [37]. Samples with boiled linseed oil surface treatment were labelled as R-MOC-LO and EPP-MOC-LO.

*2.3. Experimental Methods*

2.3.1. Aggregate Testing Methods

For the basic characterization of both used aggregates, silica sand, and crushed waste EPP (Figure 4), their specific density and grain-size distribution were measured. The thermal storage and transport parameters, as well as the powder density were also determined and these measurements were performed in the dependence on the compacting time, in order to approximate the conditions prevailing during sample preparation.

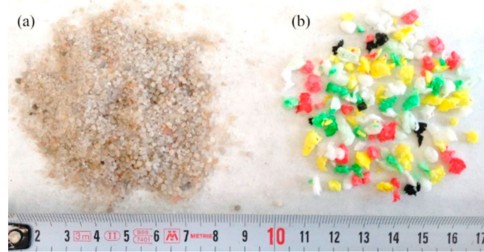

**Figure 4.** Used aggregate: silica sand (**a**); crushed waste expanded polypropylene (EPP) (**b**). Scale bar is in centimeters.

The specific density of aggregate was examined by helium pycnometry (see above). The standard sieve method using the sieves of mesh dimensions 0.063, 0.125, 0.25, 0.5, 1.0, 2.0, 4.0, 8.0, 16.0, 31.5, and 63.0 mm was used for the grain-size analysis of both aggregates. For the measurement of their thermal conductivity $\lambda$ ($W{\cdot}m^{-1}{\cdot}K^{-1}$) and volumetric heat capacity $C_v$ ($J{\cdot}m^{-3}{\cdot}K^{-1}$), the device ISOMET 2114 (Applied Precision, Ltd., Bratislava, Slovakia) equipped with a needle probe for testing granular and powder materials was applied. For experimental evaluation of the thermal parameters and powder density, the studied aggregates were inserted into the graduated cylinder. The compaction was conducted by a vibration exciter (VSB 15, Brio Hranice Ltd., Hranice, Czech Republic). From the known mass of the aggregate and its volume in a graduated cylinder, powder density was calculated.

2.3.2. MOC Cement Composites Testing Methods

The workability of the fresh composites was investigated immediately after the production of individual mixtures according to the standard EN 12350-5 [38]. For the measurement of the flow diameter, three samples of each mixture were used.

For the hardened lightweight MOC composites, basic structural characteristics, mechanical parameters, thermal properties, water- and water vapor transport parameters were investigated. SEM/EDS and XRD analyses were also conducted. The testing was carried out on a minimum of five samples of the particular studied material.

The reaction products in the MOC composites were identified by X-ray diffraction (the same parameters and device as described above). Scanning electron microscopy (SEM) with a FEG electron source (Tescan Lyra dual beam microscope) was used for the characterization of the morphology and microstructure of MOC composites. Elemental composition and mapping were performed using an energy dispersive spectroscopy (EDS) analyzer (X-Max$^N$) with a 20 mm$^2$ SDD detector (Oxford instruments) and AZtecEnergy software. To conduct the measurements, the samples were placed on a carbon conductive tape. SEM and SEM-EDS measurements were carried out using a 10 kV electron beam.

The measurements of compressive and flexural strength as well as the assessment of the dynamic Young's modulus of elasticity were conducted on 28 days cured samples. For measurement of the flexural strength, prisms with dimension of 40 mm × 40 mm × 160 mm were used. On the remains of samples from flexural strength measurement, the compressive strength was evaluated; the loading area was 40 × 40 mm.

Both strength tests were performed in accordance with the standard EN 14016-2 [35] and their relative expanded uncertainty was 1.4%. The measurement of the dynamic Young's modulus of elasticity was conducted on a pulse ultrasonic principle using an ultrasonic pulse velocity tester 58-E4800 UPV (Controls, Milan, Italy). Based on the measured wave velocity $v$ (m·s$^{-1}$) and bulk density $\rho_b$ (kg·m$^{-3}$), the dynamic Young's modulus was calculated using Equation (1):

$$E_d = \rho_b v^2 \tag{1}$$

The expanded combined uncertainty of this measurement method was 3.1%.

In order to evaluate the bulk and specific density of the hardened composites, the specimens were first dried in a vacuum drier until the changes in their mass were <0.1%. For the bulk density measurement, the test was based on a gravimetric principle according to the standard EN 12390-7 [39] and the cubic samples were employed. The expanded combined uncertainty of bulk density test was 2.4%. The specific density was accessed on the fragments from the strength tests using the helium pycnometry (see above). The expanded combined uncertainty of the specific density test was 1%. The porosity of MOC composites was then calculated using the bulk and specific density values. The relative expanded combined uncertainty of the porosity determination was 3.5%.

For the assessment of thermo-physical parameters of MOC cement composites, i.e., the thermal conductivity $\lambda$ (W·m$^{-1}$·K$^{-1}$) and the volumetric heat capacity $C_v$ (J·m$^{-3}$·K$^{-1}$), apparatus thermal constant analyzer hot disk TPS 1500 (Hot Disk AB, Gothenburg, Sweden) was applied. The apparatus works on a transient plane source method based on the use of a transiently heated plane sensor consisting of double spiral shape nickel-metal wire with diameter 10 µm embedded between two thin special foils keeping it electrically insulated. The measuring range of the thermal conductivity of TPS 1500 device was from 0.03 to 500 W·m$^{-1}$·K$^{-1}$ with the measuring accuracy ±5% and reproducibility ±2%. In case of the volumetric heat capacity, reproducibility corresponded to ±7% respectively. The measurement was made for 28 days air-cured cubic specimens with a side of 70 mm, which were first dried in a vacuum drier at 60 °C. The tests were conducted under laboratory conditions at a temperature of 23 ± 2 °C.

In order to evaluate the resistance of MOC cement composites coated with linseed oil and/or enriched with inner hydrophobic agents against water penetration, the measurement of water transport parameters was done. For the water sorptivity tests, the 28 days air-cured cubic samples with a side of 70 mm, dried in a vacuum drier at 60 °C, were used. The assessment of water absorption coefficient $A$ (kg·m$^{-2}$·s$^{-1/2}$) and water sorptivity $S$ (m·s$^{-1/2}$) was conducted using the free water intake test [40]. All lateral sides of the samples were first water- and vapor-proof insulated by epoxy resin. Their face sides, 70 mm × 70 mm, were then immersed in the distilled water and the weight gain at the chosen time intervals was registered. From the slope of the initial stage of the cumulative mass of water plotted vs. square root of time, the water absorption coefficient $A$ was determined. The water sorptivity $S$ was then calculated from the equation:

$$A = S \cdot \rho_w \tag{2}$$

where $\rho_w$ (kg·m$^{-3}$) is the density of water. The expanded combined uncertainty in the determination of the water absorption coefficient and sorptivity was 2.3%.

The characterization of MOC cement composites with and without inner and surface hydrophobic treatments in terms of water vapor transmission parameters was done using the dry-cup method following the standard EN ISO 12572 [41]. The measurement was based on 1-D water vapor diffusion, where the gradient of water vapor pressure in the air above and under specific surface of sample caused the water vapor flow through the sample. The test was done in steady state under isothermal conditions (temperature of 23 ± 0.5 °C and relative humidity of 50 ± 5%). The lateral sides of cylindrical samples (diameter of 100 mm and thickness of 20 mm) were water and vapor-proof insulated using the epoxy resin so that only one-dimensional water vapor was realized. Samples were placed and sealed on the top of the cup, in which the presence of silica gel ensured the equilibrium relative humidity

of ~2% below the sample. The experiment was conducted in a controlled climatic chamber and the cups were periodically weighed to achieve constant mass gain. On the basis of the measured mass gains in time, water vapor diffusion coefficient $D$ (m$^2$·s$^{-1}$) and vapor diffusion resistance factor $\mu$ (−) were calculated.

## 3. Results and Discussion

The results of the grain-size analysis of silica sand and EPP aggregate are illustrated in Figure 5. According to the grain size curves, the EPP grains were found smaller than 8 mm and diameter of silica sand particles ranged from 0.063 μm to 2 μm. As further processing of EPP to obtain finer particles was refused in order to ensure low cost and low energy embedded in raw materials, the granulometry of EPP was considered as good in general for substitution of sand in composition of MOC cement composites.

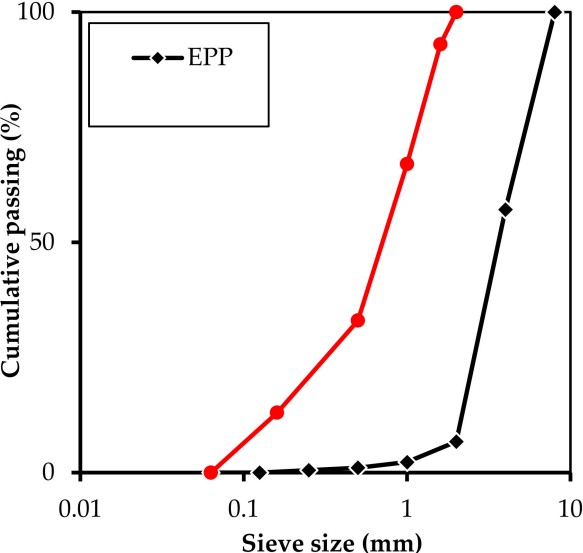

**Figure 5.** Granulometric curves of used silica sand and EPP particles. Sieve size is given in logarithmic scale.

For the reduction of the thermal conductivity values of the tested MOC cement composites it was desirable to select the filler having a low thermal conductivity compared to silica sand commonly used in the construction industry. Thermal and physical properties of both investigated aggregates are summarized in Table 4.

**Table 4.** Thermo-physical parameters of used aggregates (influence of time of compaction).

| Aggregate | Specific Density (kg·m$^{-3}$) | Time of Compaction (s) | Powder Density (kg·m$^{-3}$) | $\lambda$ (W·m$^{-1}$·K$^{-1}$) | $C_v$ (× 10$^6$ J·m$^{-3}$·K$^{-1}$) |
|---|---|---|---|---|---|
| EPP | 105 | 0 | 18.9 | 0.042 | 0.049 |
| | | 10 | 22.4 | 0.043 | 0.055 |
| | | 30 | 23.3 | 0.044 | 0.058 |
| | | 60 | 23.9 | 0.044 | 0.061 |
| | | 180 | 24.3 | 0.047 | 0.062 |
| Silica sand | 2652 | 0 | 1657 | 0.410 | 1.569 |
| | | 10 | 1910 | 0.562 | 1.681 |
| | | 20 | 1916 | 0.575 | 1.683 |
| | | 30 | 1922 | 0.576 | 1.684 |
| | | 60 | 1927 | 0.577 | 1.689 |

The longer the time of compaction, the higher thermal conductivity and volumetric heat capacity values were measured. It was due to the increasing packing density with compaction and reduction of

air volume between the aggregate particles. As expected, the EPP particles in comparison with those of silica sand exhibited significantly lower heat transport and heat storage. It corresponded with their low specific density, chemical origin, and structure. Nevertheless, for the practical purposes, the low thermal conductivity of EPP is beneficial for the production of lightweight MOC cement composites with an improved thermal insulation function. In Table 5, the workability of fresh mixtures expressed in terms of flow diameter is given.

**Table 5.** Diameter of fresh mixtures.

| Material | R-MOC | EPP-MOC | EPP-MOC-IH |
|---|---|---|---|
| Flow diameter (mm) | 229 | 145 | 142 |

Generally, the workability of fresh mixtures affects particularly the amount and size of used filler. The shape of the particles also plays a role [32,42]. Results showed that mixtures containing EPP with the higher particle size and expected higher specific surface compared to those of silica sand exhibited considerably reduced workability than the control mixture with silica filler. However, for the intended application of MOC cement composites in thermal insulation floor layers or prefabricated panels and slab with EPP was the achieved workability quite sufficient. The workability of mixtures EPP-MOC and EPP-MOC-IH was almost the same, hydrophobic agents had no clear effect on the flow.

In Table 6, the structural parameters of the developed MOC cement composites are summarized. The observed R-MOC bulk density value of 2130 kg·m$^{-3}$ was similar as the dry density of 2142 kg·m$^{-3}$ reported for reference MOC composite by Xu et al. [12]. With the addition of the EPP aggregate, the bulk density of composites significantly decreased to 42.6% and 43.5% for EPP-MOC and EPP-MOC-IH respectively, compared to R-MOC. It can be explained by the lower specific density of EPP particles (see Table 4). According to EN 206-1 [43], the developed EPP-MOC and EPP-MOC-IH may be classified based on their bulk densities as lightweight concretes in class LC 1.0. The porosity of MOC cement composites containing EPP aggregate increased compared to R-MOC. Similar behavior can be found in literature regarding the incorporation of different kind of plastics into a matrix based on Portland cement [32,44]. The lightening of high-density structure by the use of the EPP aggregate resulted in lightweight composites with high porosity which extended their application potential in the construction sector. The used hydrophobic admixtures had almost no effect on structural characteristics of composites with EPP.

**Table 6.** Basic structural characteristics of lightweight MOC cement composites.

| Material | Specific Density (kg·m$^{-3}$) | Bulk Density (kg·m$^{-3}$) | Porosity (%) |
|---|---|---|---|
| R-MOC | 2460 ± 25 | 2130 ± 51 | 13.4 ± 0.5 |
| EPP-MOC | 1420 ± 14 | 908 ± 22 | 36.1 ± 1.2 |
| EPP-MOC-IH | 1413 ± 14 | 926 ± 22 | 34.5 ± 1.2 |

The comparison of strength characteristics of examined composites is given in Table 7.

**Table 7.** Mechanical resistance of MOC composites.

| Material | Compressive Strength (MPa) | Flexural Strength (MPa) | Young´s Modulus (GPa) |
|---|---|---|---|
| R-MOC | 63.2 ± 0.9 | 19.3 ± 0.3 | 36.6 ± 1.1 |
| EPP-MOC | 6.3 ± 0.1 | 4.8 ± 0.1 | 4.0 ± 0.1 |
| EPP-MOC-IH | 7.6 ± 0.1 | 3.0 ± 0.0 | 5.0 ± 0.2 |

Mechanical parameters of MOC composites are mainly dependent on used molar ratios of MgO/MgCl$_2$ and H$_2$O/MgCl$_2$, which governs the formation of binding phases and amount of unreacted raw materials [45]. For mortars, the parameters of filler and ratio of filler/binder should be moreover

considered. The replacement of sand with EPP aggregate led to the significant reduction in all strength parameters. The compressive strength reductions of EPP-MOC and EPP-MOC-IH were 90% and 88%, respectively, as compared to control material R-MOC. This strength decrease is in accordance with the density and porosity data (Table 6). Similarly as in our case, many researchers stated that the incorporation of plastics aggregate of varying shape, kind, and amount causes the loss of mechanical resistance in the composites based on the Portland cement [30,32,45]. Accordingly, the decrease of compressive strength of MOC cements containing different additives was reported by Zgueb [33].

The typically higher flexural/compressive strength ratio of MOC composites compared to that based on the Portland cement was observed for all developed MOC samples. It can be anticipated that the use of smaller EPP particles would increase the mechanical resistance of the final composites. The improvement in strength parameters might by possible also by vibrating the fresh casted specimens.

The decrease in the thermal transport and storage parameters of MOC composites with EPP aggregate content is obvious from Table 8.

**Table 8.** Thermal performance of lightweight MOC composites.

| Material | $\lambda$ (W·m$^{-1}$·K$^{-1}$) | SD | $C_v$ ($\times 10^6$ J·m$^{-3}$·K$^{-1}$) | SD |
|---|---|---|---|---|
| R-MOC | 2.09 | 0.007 | 1.75 | 0.03 |
| EPP-MOC | 0.34 | 0.009 | 1.53 | 0.03 |
| EPP-MOC-IH | 0.35 | 0.001 | 1.55 | 0.03 |

SD—standard deviation.

The great drop in the thermal conductivity values noticed for materials EPP-MOC and EPP-MOC-IH were 83.7% and 83.3%, respectively, compared to R-MOC. In the case of the volumetric heat capacity, a decrease of ~12% was observed for MOC composites containing EPP. Xu et al. [12] reported on the reduction of thermal conductivity of MOC-based composites with addition of cenospheres derived from fly ash. For cenospheres, addition in the amount of 5, 15, and 25% by weight of the magnesia powder, they observed the thermal conductivity values reduced by 16, 21 and 32%, respectively, as compared to the reference sample without cenospheres. In our case, the deceleration of heat transport by the use of the substitution of silica sand with crushed EPP was much higher. In summary, the main parameters that affect the resulted thermal properties of developed MOC composites were the amount and type of used aggregates, thermal parameters of the aggregates themselves, and the porosity of the composites which significantly increased because of the use of EPP particles. One should take into account the fact that the obtained thermo-physical parameters were measured on dried samples only and the exposure of the studied materials to the real operations conditions of buildings will partially reduce their thermal insulation capability because of the presence of adsorbed water vapor molecules.

The resulting values of water transport properties are shown in Table 9. In the water sorptivity test, samples with inner (IH) and surface (LO) hydrophobic treatment were examined. Unfortunately, the water suction experiment was not conducted for control R-MOC composite without any hydrophobic treatment because the samples in contact with water deteriorated by the excessive moisture presence. As the specimens exhibited volume changes and cracking, it was not possible to evaluate the measured data.

**Table 9.** Water transport properties of MOC composites with inner and surface hydrophobic treatment.

| Material | $A$ (kg·m$^{-2}$·s$^{-1/2}$) | $S$ (m$^2$·s$^{-1/2}$) |
|---|---|---|
| R-MOC-LO | $0.0002 \pm 5 \times 10^{-6}$ | $2.00 \times 10^{-7} \pm 5 \times 10^{-9}$ |
| EPP-MOC-LO | $0.0007 \pm 2 \times 10^{-5}$ | $6.51 \times 10^{-7} \pm 2 \times 10^{-8}$ |
| EPP-MOC-IH | $0.0014 \pm 3 \times 10^{-5}$ | $1.35 \times 10^{-6} \pm 3 \times 10^{-8}$ |

The values of liquid water transport properties for all tested materials were very low. The reference MOC composite had the lowest water absorption among the samples tested. The increased porosity of the MOC composites with EPP (Table 6) resulted in a higher water absorption coefficient and water sorptivity. Comparing EPP-MOC samples having inner and surface hydrophobic treatment, the water absorption coefficient was lower for samples treated with boiled linseed oil. However, inner hydrophobic agent provided water-repellent effect both on the surface and in the entire material treated.

No data on hygric properties of MOC cement composites for comparison of our results were found in common literature sources. Generally, hydrophobic treatment significantly affects the water transport properties. For the Portland cement-based composites having porosity of 14.0%, which is similar to R-MOC sample (13.4%), Záleská et al. [32] reported on the water absorption coefficient of $0.0347 \, \text{kg} \cdot \text{m}^{-2} \cdot \text{s}^{-1/2}$ which was more than two orders of magnitude higher than measured for R-MOC-LO. This comparison clearly validated high effectiveness of both inner and surface hydrophobization. Based on that high resistance to moisture attack can be estimated for MOC cement composites treated with the tested chemical additives.

Table 10 shows the water vapor transmission parameters of the investigated MOC cement composites. The water vapor transmission experiments were done on samples without any hydrophobic treatment and on samples with inner and surface hydrophobic treatment. Comparing the MOC composites without hydrophobisation, the water vapor diffusion coefficient of EPP-MOC was about 10% higher than that of the R-MOC sample. The surface treatment with boiled linseed oil reduced the water vapor diffusion coefficient by almost 43% and 12% for reference sample and sample with EPP, respectively. In the case of the inner hydrophobic treatment, EPP-MOC and EPP-MOC-IH exhibited very similar water vapor transmission parameters which pointed out the fact that the inner hydrophobisation predominantly affected the transport of liquid water and not the water vapor properties. This finding is crucial for practical use of the developed lightweight composites as the permeability for water vapor eliminates the possible condensation problems and health risks of the indoor environment, such as respiratory diseases, allergies, etc.

**Table 10.** Water vapor transport properties of MOC composites.

| Material | $D \, (\text{m}^2 \cdot \text{s}^{-1})$ | $\mu \, (-)$ |
|---|---|---|
| R-MOC | $5.46 \times 10^{-7}$ | 46.0 |
| R-MOC-LO | $3.14 \times 10^{-7}$ | 78.9 |
| EPP-MOC | $6.47 \times 10^{-7}$ | 38.3 |
| EPP-MOC-LO | $5.72 \times 10^{-7}$ | 43.3 |
| EPP-MOC-IH | $6.40 \times 10^{-7}$ | 38.7 |

The phase composition of prepared composites was measured using XRD (see Figure 6). Presence of MOC phase (ICDD 00-007-0420) was confirmed in all three samples. The reaction between $MgCl_2$ and MgO followed the equation below:

$$5 \, MgO + MgCl_2 \cdot 6H_2O + 3 \, H_2O \rightarrow 2 \, [Mg_3(OH)_5Cl \cdot 4H_2O] \tag{3}$$

All samples contained over-stoichiometry of MgO (ICDD 01-075-1525), hence MgO was present in all samples after the complete depletion of magnesium chloride in the reaction mixture. This was expected as MgO was used partially as filler. Sample R-MOC contained mainly quartz (ICDD 01-083-2471) and MOC phase and some residual MgO. Sample R-MOC contained mainly quartz and MOC phase. Also some residues of MgO were present. Sample EPP-MOC-IH contained two major phases: MOC and MgO. The last sample EPP-MOC contained mainly MOC phase and only very low amount of MgO.

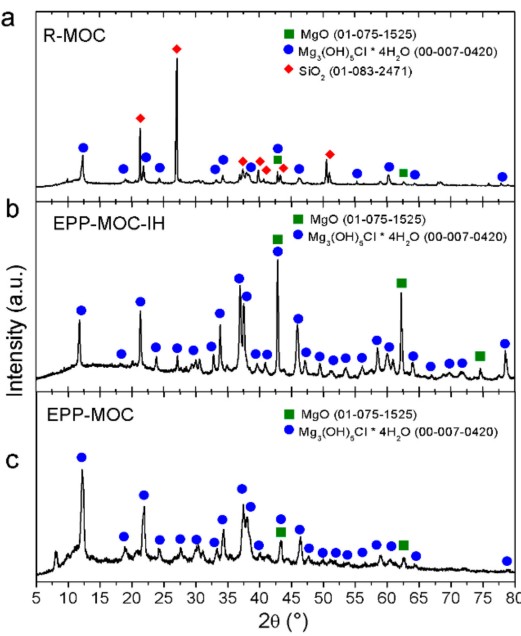

**Figure 6.** Diffractograms of R-MOC (**a**); EPP-MOC-IH (**b**); EPP-MOC (**c**).

The microstructure of prepared composites was measured using SEM and the results are shown in Figure 7. In all samples, the needle-like structures typical for MOC phase were detected. The boundaries between the matrix and EPP (samples EPP-MOC-IH and EPP-MOC) are highly compact without cracks or other defects. Selected areas of composites were also analyzed using EDS. While the chemical composition varies significantly from region to region, the elemental composition values are not shown here. However, according to elemental maps the grain boundaries between EPP and MOC matrix are clearly visible.

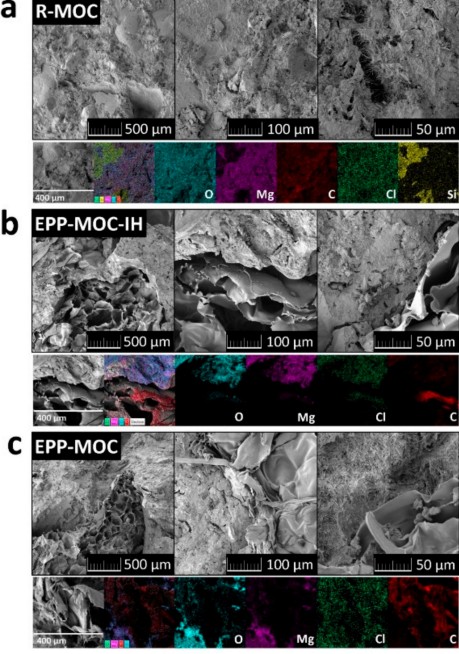

**Figure 7.** SEM micrographs and elemental maps of R-MOC (**a**); EPP-MOC-IH (**b**); EPP-MOC (**c**).

## 4. Conclusions

In order to meet increasing energy efficiency standards for buildings and reduce $CO_2$ footprint of construction industry, researches are looking for alternatives to common building materials. Such types of "green materials" were developed and tested in the presented study. The use of low-energy and low-carbon MOC cement and application of waste EPP aggregate in composition of new types of alternative construction composites met present criteria of sustainable development. The developed lightweight MOC cement composites exhibited improved thermal insulation function, minimum water absorption, and sufficient permeability for water vapor. The produced materials also showed good water resistance, which is the biggest problem of common MOC cement-based materials. The mechanical strength of the lightened composites was acceptable for non-bearing purposes, but it can be further improved by the use of EPP aggregate with smaller particle size and by vibration of the freshly casted specimens in molds. These experiments and tests will be conducted in our future studies together with the testing of acoustic properties whose improvement by the use of EPP aggregate can be also anticipated.

It can be concluded that the use of waste EPP in combination with MOC cement and hydrophobic agents made it possible to develop advanced types of building composites with added value, such as eco-efficiency, low cost, low embodied energy, and durability.

**Author Contributions:** Conceptualization, M.P. and Z.P.; methodology, M.P., O.J., and Z.P.; investigation, M.Z., O.J., M.L., F.A., A.P.; data curation, M.P., O.J., and Z.P., writing—original draft preparation, M.Z., M.P., and Z.P.; supervision, M.P.; project administration, O.J., M.P.

**Funding:** This research was funded by the CZECH SCIENCE FOUNDATION, grant number 19-00262S—Reactive magnesia cements-based composites with selected admixtures and additives.

**Acknowledgments:** The technical support provided by Pavel Košata from the Faculty of Civil Engineering, CTU Prague, is greatly acknowledged.

**Conflicts of Interest:** The authors declare no conflict of interest.

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
