# Peer review of "Influence of Waste Plastic Aggregate and Water-Repellent Additive on the Properties of Lightweight Magnesium Oxychloride Cement Composite"

_applsci, doi:10.3390/app9245463_

Round 1

Reviewer 1 Report

Authors of the article entitled "Influence of Waste Plastic Aggregate and Water-Repellent Additive on the Properties of Lightweight Magnesium Oxychloride Cement Composite" focused mainly on the physical and thermal parameters of the MOC cements with EPP. The article is well structured and the results are properly discussed, but some elements need to be improved and completed:

- Abstract- should be shortened; first three sentences are more suitable for the introduction part, not the abstract;

- subsection 2.1 'Materials and Methods'- this title is confusing because usually in such titled subsection all the used materials and methods are described and here these information are divided into subsection 2.1 and 2.2; the best option will be the change of the title of the subsection 2.1 to be more suitable for the included information;

- p.7, line 255- bracket should be removed;

- p.12, lines 402-406- the information contained in Figure 5 should be further discussed, you describe the diffraction patterns/products but there is no explanation of the chemical mechanism of the reaction products. It should be more explained.

Author Response

We highly appreciate reviewers’ and editor’s work on review of our manuscript. We addressed all the reviewers’ comments and followed reviewers’ suggestions in the revised version of the manuscript. We believe that reviewers’ comments and suggestions have contributed substantially to the improvement of the presentation of our study and the overall quality of the manuscript. We performed also final reading, checked references, cleaned typos we found and grammatically corrected the revised manuscript. The revised text is in the manuscript marked up by a red colour and the particular changes are described in comments bellow.

Reviewer 1

Authors of the article entitled "Influence of Waste Plastic Aggregate and Water-Repellent Additive on the Properties of Lightweight Magnesium Oxychloride Cement Composite" focused mainly on the physical and thermal parameters of the MOC cements with EPP. The article is well structured and the results are properly discussed, but some elements need to be improved and completed:

Thank you for such positive review of our paper. We addressed all your comments and suggestions that help us improve the overall quality of our manuscript. The particular comments are responded bellow.

- Abstract- should be shortened; first three sentences are more suitable for the introduction part, not the abstract;

We have shortened Abstract as recommended.

- subsection 2.1 'Materials and Methods'- this title is confusing because usually in such titled subsection all the used materials and methods are described and here these information are divided into subsection 2.1 and 2.2; the best option will be the change of the title of the subsection 2.1 to be more suitable for the included information;

Thank you for your suggestion. We have changed the title of subsection 2.1. The new title is 2.1 Materials. We believe this title is more suitable for the included information as this section describes all the materials used in our study.

- p.7, line 255- bracket should be removed;¨

The text was revised as suggested. The brackets were removed from the whole manuscript.

- p.12, lines 402-406- the information contained in Figure 5 should be further discussed, you describe the diffraction patterns/products but there is no explanation of the chemical mechanism of the reaction products. It should be more explained.

We agree with you, the information given in Figure 5 was incompletely discussed. We have completed information on the chemical mechanisms of the precipitation of reaction products and presence of unreacted MgO that acted partially as filler.

Reviewer 2 Report

Measurement results of the parameters presented in the article should be extended with basic statistical information, e.g. standard deviation.

Among the advantages of the tested composite, the authors mention its durability. Was it the subject of aging tests ((freezing tests, UV tests etc.) or was based only on the results of the research presented in the paper?

A composite with a heat transfer coefficient of 0.3 W / mK has a negligible effect on the thermal resistance of partitions separating environments at different temperatures, which is why in future I suggest developing research into acoustic and strength properties

Author Response

Thank you for such positive review of our paper. We addressed all your comments and suggestions that help us improve the overall quality of our manuscript. The particular comments are responded bellow.

Measurement results of the parameters presented in the article should be extended with basic statistical information, e.g. standard deviation.

We agree with you, basic statistical information on measured data is important. As most of the methods used for the characterization of hardened composites are in our laboratory accredited according to EN ISO/IEC 17025:2018 standard, we are used to add information on measurement uncertainty, in this case expanded combined uncertainty which gives 95% confidence level. Information on measurement uncertainty related to the measured data was completed in Tables 6, 7, 8, 9, 10.

Among the advantages of the tested composite, the authors mention its durability. Was it the subject of aging tests ((freezing tests, UV tests etc.) or was based only on the results of the research presented in the paper?

Yes, we have only estimated the durability of the developed composites based on their minimum water absorption which thus limits harmful freeze/thaw effects.

A composite with a heat transfer coefficient of 0.3 W / mK has a negligible effect on the thermal resistance of partitions separating environments at different temperatures, which is why in future I suggest developing research into acoustic and strength properties

Thank you for your idea of our future research. It is highly appreciated. The aim of our future research was extended by the investigation of acoustic properties of composites with EPP aggregate.